# An open-source tool for automated analysis of breathing behaviors in common marmosets and rodents

**Mitchell Bishop, Maximilian Weinhold, Ariana Z Turk, Afuh Adeck, Shahriar SheikhBahaei***

Neuron-Glia Signaling and Circuits Unit, National Institute of Neurological Disorders and Stroke (NINDS), National Institutes of Health (NIH), Bethesda, United States

**Abstract** The respiratory system maintains homeostatic levels of oxygen ($O_2$) and carbon dioxide ($CO_2$) in the body through rapid and efficient regulation of breathing frequency and depth (tidal volume). The commonly used methods of analyzing breathing data in behaving experimental animals are usually subjective, laborious, and time-consuming. To overcome these hurdles, we optimized an analysis toolkit for the unsupervised study of respiratory activities in animal subjects. Using this tool, we analyzed breathing behaviors of the common marmoset (*Callithrix jacchus*), a New World non-human primate model. Using whole-body plethysmography in room air as well as acute hypoxic (10% $O_2$) and hypercapnic (6% $CO_2$) conditions, we describe breathing behaviors in awake, freely behaving marmosets. Our data indicate that marmosets' exposure to acute hypoxia decreased metabolic rate and increased sigh rate. However, the hypoxic condition did not augment ventilation. Hypercapnia, on the other hand, increased both the frequency and depth (i.e., tidal volume) of breathing.

## Editor's evaluation

The authors have thoughtfully revised their manuscript, with an increased focus on their breath analysis toolkit. We think this will be a tremendous resource for the respiratory community and we are hopeful it will decrease the barriers for others to conduct similar investigations.

**\*For correspondence:**
SheikhbahaeiS@nih.gov

**Competing interest:** The authors declare that no competing interests exist.

## Introduction

Mammals rely on a continuous supply of oxygen ($O_2$) from the environment and efficient removal of carbon dioxide ($CO_2$) and other metabolic waste products from their body. The intricate respiratory system ensures the homeostatic state of the arterial partial pressure of $O_2$ ($PO_2$) and $CO_2$ ($PCO_2$) in the blood by executing rhythmic movement of the respiratory pump, which includes the intercostals and the diaphragm muscles (*Del Negro et al., 2018*). The inception of this respiratory rhythm occurs within the preBötzinger complex (preBötC), a functionally specialized region in the ventrolateral medulla of the brainstem (*Smith et al., 1991*; *Del Negro et al., 2018*). Activities of the preBötC are modulated by specialized peripheral and central chemosensors that adjust the respiratory drive to regulate homeostatic levels of $PO_2$ and $PCO_2$ (*Heymans and Bouckaert, 1930*; *O'Regan and Majcherczyk, 1982*; *Guyenet, 2014*; *Sheikhbahaei et al., 2018*; *Angelova et al., 2015*; *van der Heijden and Zoghbi, 2020*; *Guyenet et al., 2019*; *Sheikhbahaei et al., 2017*; *Del Negro et al., 2018*).

Most studies on homeostatic control of breathing have been done in rodent models, in which the experiments are mostly performed during the day, rodents' normal inactive period. Since, in general, rodents have relatively reduced chemosensitivities compared with primates (*Hazari and Farraj,*

*2015*), the use of non-human primates (NHPs) has been proposed to fill the gap and translate rodent breathing data to humans (*SheikhBahaei, 2020*). The common marmoset (*Callithrix jacchus*) is a New World NHP with a small body size (250–600 g) similar to that of a rat. Ease of handling, high reproductive efficacy, and lack of zoonotic risks compared to Old World NHPs make marmosets an attractive and powerful NHP model for biomedical and neuroscience research (*Abbott et al., 2003*). Marmosets have been proposed as a primate model to study behavioral neuroscience, leading to a recent increase of their use in research settings (*Prins et al., 2017*; *Miller et al., 2016*; *Walker et al., 2017*). However, the basic characteristics of breathing behaviors in marmosets are not yet defined.

Whole-body plethysmography has been widely used in studying breathing behaviors in animal models (*Besch et al., 1996*; *Iizuka et al., 2010*; *Sheikhbahaei et al., 2018*; *Hosford et al., 2020*; *Liu et al., 2016*; *Hutchison et al., 1983*; *Hoffman et al., 1999*; *Valente et al., 2012*; *Tattersall et al., 2002*). However, analyzing whole-body respiratory data in awake animals requires algorithms to distinguish different respiratory signals. To avoid this problem, respiratory activities are often recorded when the animal is asleep, awake with minimal movement, or anesthetized. Yet studying the homeostatic control of breathing physiology in awake animals has absolute advantages, despite the increased variability. Therefore, to overcome this challenge, we developed an open-source Python tool using Neurokit2 (*Makowski et al., 2021*), for unsupervised analysis of respiratory signals obtained from rats and common marmosets. We then characterized the ventilatory responses in marmosets at rest as well as during acute hypoxia (decrease inspired $O_2$ to 10%) and hypercapnia (increased inspired $CO_2$ to 6%). We found that while exposure of marmosets to hypoxia increased sigh rate and decreased overall animal metabolic rate, the hypoxia-induced augmentation of ventilation was diminished. On the other hand, hypercapnic conditions increased both frequency and depth of breathing.

## Results

### Validation of the analysis toolkit in experimental animal models

Analysis of breathing data from plethysmography is usually time-consuming, laborious, and often involves measurements of rate of breathing ($f_R$), tidal volume ($V_T$), and minute ventilation ($V_E$). $f_R$ is

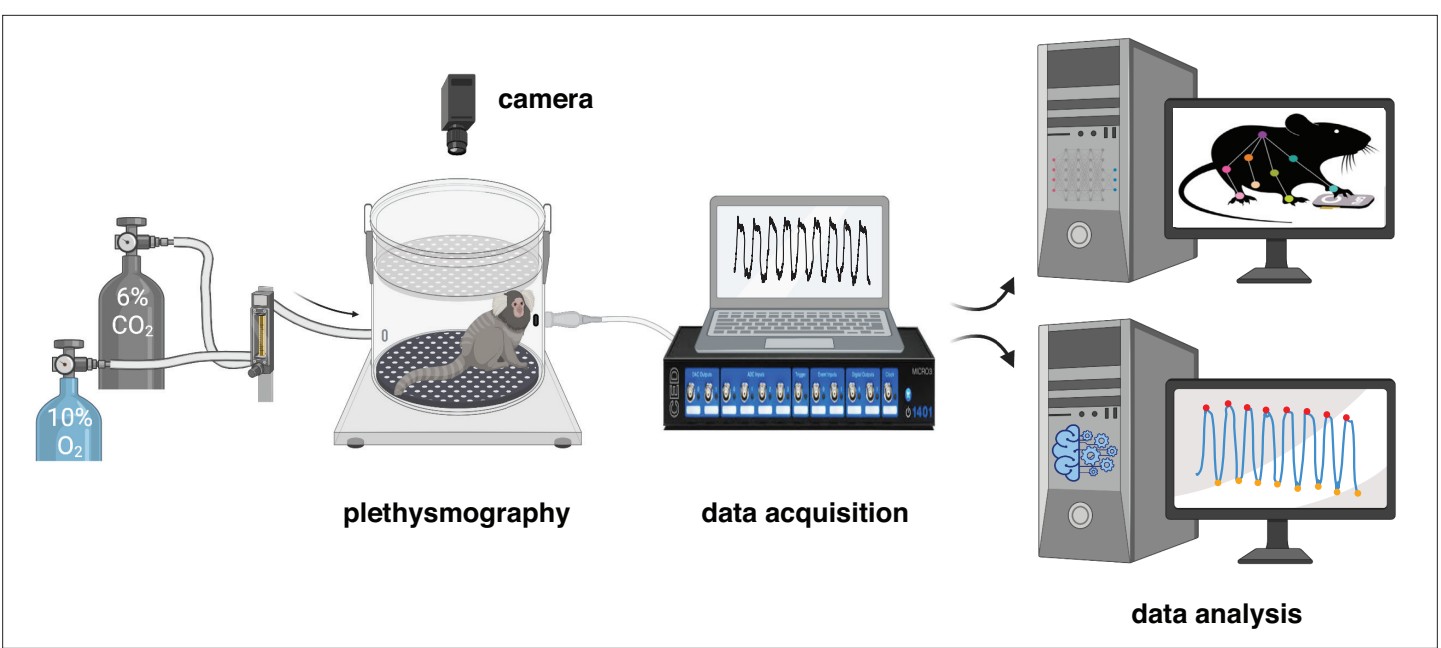

**Figure 1.** Experimental design for measurement and analysis of marmoset respiratory behaviors. After a 40 min baseline period at room air (21% $O_2$, ~0% $CO_2$, and 79% $N_2$), the breathing behavior of the animal was studied under either hypoxic (10% $O_2$; 10 min) or hypercapnic (6% $CO_2$; 10 min) conditions. Raw respiratory signal is later cleaned and analyzed offline (see Materials and methods for details). Video of spontaneous activity in the chamber at baseline and during each challenge was used to train a DeepLabCut model to track the animal body.

The online version of this article includes the following figure supplement(s) for figure 1:

**Figure supplement 1.** Breathing behaviors in adult marmoset.

usually calculated in intervals when the animal is asleep or stationary from the time of peak-to-peak inspiratory or expiratory signal. $V_T$ is often measured by integration of signal over a specified period of time. Therefore, analysis of minute-to-minute changes in breathing may be difficult and take more time. In addition, the fact that conventional analysis is usually subjective might affect the reproducibility of reported results. To overcome this problem, we wrote a custom, open-source Python script using Neurokit2, NumPy, and Pandas software packages (*McKinney, 2010*; *van der Walt et al., 2011*; *Makowski et al., 2020*) to analyze breathing signals in awake, freely-moving animals (*Figure 1*). While using this script, the user has the option to define the start and end intervals for baseline as well as experimental challenges, which the script uses to import the data and perform analysis. To validate our script, we benchmarked the data analyzed against a conventional method (*Sheikhbahaei et al., 2018*; *Sheikhbahaei et al., 2017*) by analyzing simple respiratory data ($f_R$, $V_T$, and $V_E$) in conscious marmosets and rats (*Figure 2*). We did not identify any differences in values of $f_R$, $V_T$, and $V_E$ between those generated using our script or by the conventional method (n = 3 per species) (*Figure 2—figure supplement 1* and *Figure 2—figure supplement 2*). We then used our toolkit to further analyze other breathing behaviors in both male and female marmosets at room air and during acute exposure to hypoxia and hypercapnia (see below).

## Resting respiratory behavior in adult marmosets

The $f_R$ at room air (normoxia/normocapnia) was similar in female (79 ± 7 breaths min$^{-1}$, n = 8) and male (78 ± 8 breaths min$^{-1}$, n = 8) adult marmosets (p = 0.88, Mann–Whitney test) (*Figure 2—figure supplement 3*). The $V_T$, calculated from trough to peak amplitude and normalized to body mass, was similar in female (0.43 ± .10 a.u.) and male (0.53 ± 0.08 a.u.) adult marmosets as well (p = 0.37, Mann–Whitney test). Additionally, baseline $V_E$ was similar in female (35 ± 10 a.u.) and male (42 ± 8 a.u.) marmosets (p = 0.38, Mann–Whitney test) (*Figure 2—figure supplement 3*). Two marmosets (one male and one female) showed prolonged breath holding (11 ± 2 breaths hr$^{-1}$ for 4.3 ± 0.1 s).

## Ventilatory response to acute hypercapnia in adult marmosets

We also measured changes in $f_R$, $V_T$, and $V_E$ before, during, and after acute hypercapnic challenge (6% $CO_2$ in the inspired air). The magnitude of change in $f_R$, $V_T$, and $V_E$ was similar between females and males during hypercapnia (n = 4 per sex, *Figure 3—figure supplement 1*), so we grouped them for further analyses. Increasing $CO_2$ inside the chamber increased $f_R$ (87 ± 8 vs. 74 ± 8 breaths min$^{-1}$ in baseline, p = 0.039, Wilcoxon matched-pairs signed rank test), $V_T$ (1.04 ± .11 vs. 0.4 ± 0.05 a.u. in baseline, p = 0.008, Wilcoxon matched-pairs signed rank test) and $V_E$ (81 ± 11 vs. 32 ± 5 a.u. in baseline, p = 0.008, Wilcoxon matched-pairs signed rank test) (*Figure 3*).

Hypercapnic-induced increase in $f_R$ was mainly due to decrease in time of inspiration ($T_I$) (0.26 ± 0.02 vs. 0.36 ± 0.03 s at baseline, p = 0.008, Wilcoxon matched-pairs signed rank test) rather than time of expiration ($T_E$) (0.48 ± 0.06 vs. 0.58 ± 0.09 at baseline sec, p = 0.078, Wilcoxon matched-pairs signed rank test). As expected, respiratory flow ($R_F$) was also increased during hypercapnia (4.1 ± 0.4 vs. 1.2 ± 0.2 a.u. in baseline, p = 0.008, Wilcoxon matched-pairs signed rank test) (*Figure 3*, *Figure 3—figure supplement 2*).

Subsequently, we measured regularity of respiration via cycle-to-cycle dispersion of $T_{TOT}$ in baseline and hypercapnic condition as shown in Poincaré plots (*Figure 3*). We quantified the regularity of breathing (*Sheikhbahaei et al., 2017*) by SD1 and SD2 (see Materials and methods and *Soni and Muniyandi, 2019*). The baselines SD1 and SD2 were greater than those during hypercapnia (132 ± 17 vs. 550 ± 115 a.u. in baseline, p = 0.008, and 198 ± 34 vs. 758 ± 138 in baseline, p = 0.008, respectively; Wilcoxon matched-pairs signed rank test) (*Figure 3* and *Figure 3—figure supplement 3*).

## Ventilatory response to acute hypoxia in adult marmosets

We then measured changes of $f_R$, $V_T$, and $V_E$ during acute systemic hypoxic challenges (10% $O_2$ in the inspired air) with respect to the baseline. Similar to hypoxia, the magnitude of the change in $f_R$, $V_T$, and $V_E$ was not different in females and males during acute hypoxia (n = 4 per sex) (*Figure 4—figure supplement 1*), therefore we combined all the data from both sexes. In the first minute of the hypoxic challenge, $V_T$ and $V_E$ increased by 17% ± 12% and 17% ± 14%, respectively (*Figure 4*). This initial increase in ventilation may be due to hypoxic-induced carotid body activation. We then analyzed breathing behaviors 5 min after changing the inspired $O_2$ from 21% (room air) to 10%. Hypoxic

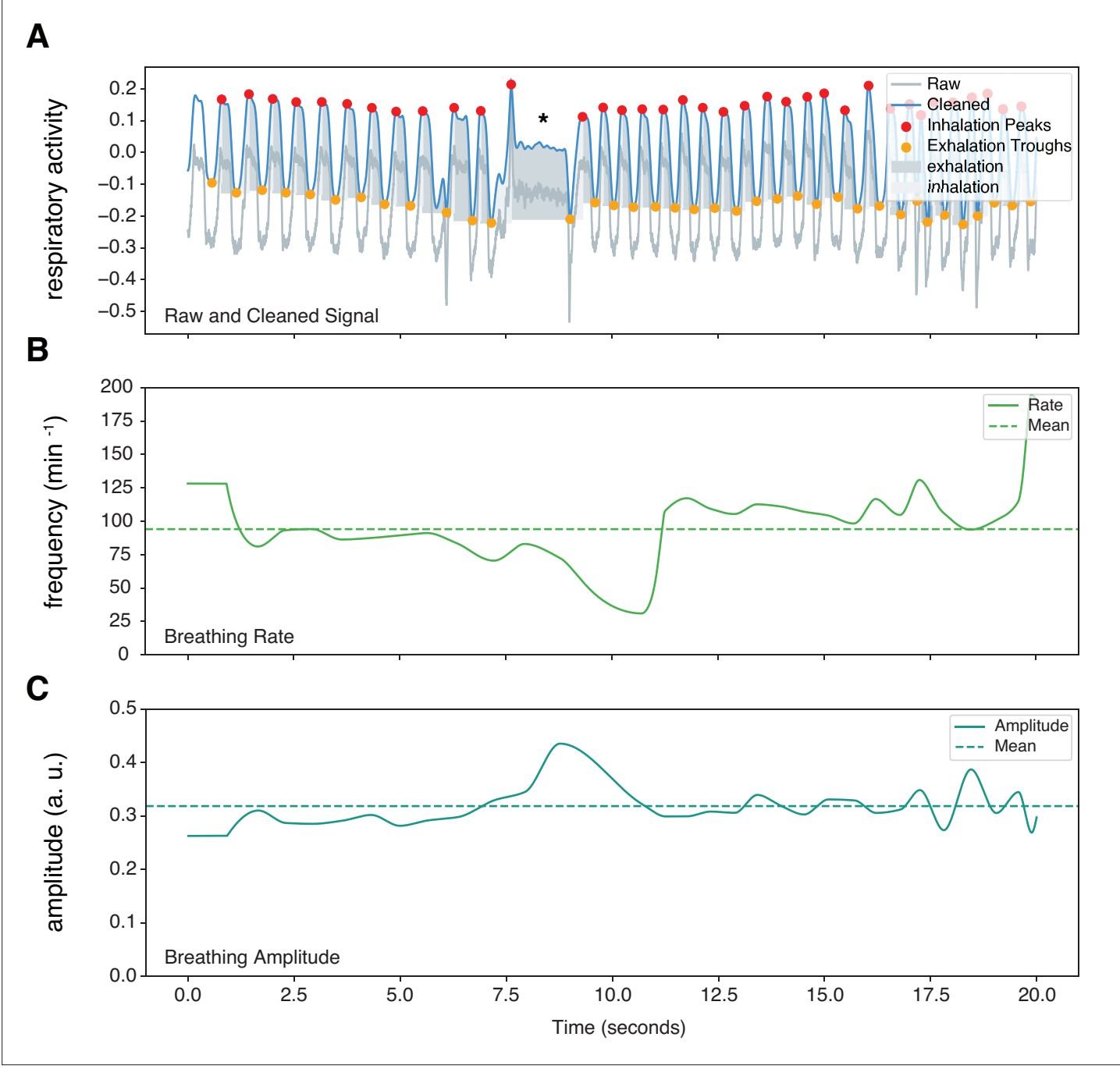

**Figure 2.** Sample marmoset respiratory trace output from Neurokit2. Representative respiratory trace is sampled from a single male marmoset during hypercapnia challenge. (**A**) NeuroKit2 was used for signal detrending and smoothing, peak and trough extraction, as well as respiratory phase. (B and C) Instantaneous measurement of breathing frequency ($f_R$) (**B**) and breathing amplitude ($V_T$) (**C**) are illustrated. This sample also contained respiratory changes during a *phee* call (marked by *). a. u. – arbitrary unit.

The online version of this article includes the following figure supplement(s) for figure 2:

**Figure supplement 1.** Validation of our tool in analysis of breathing behaviors in common marmoset.

**Figure supplement 2.** Analysis of rodent respiratory behaviors.

**Figure supplement 3.** Sex differences in baseline respiratory frequencies.

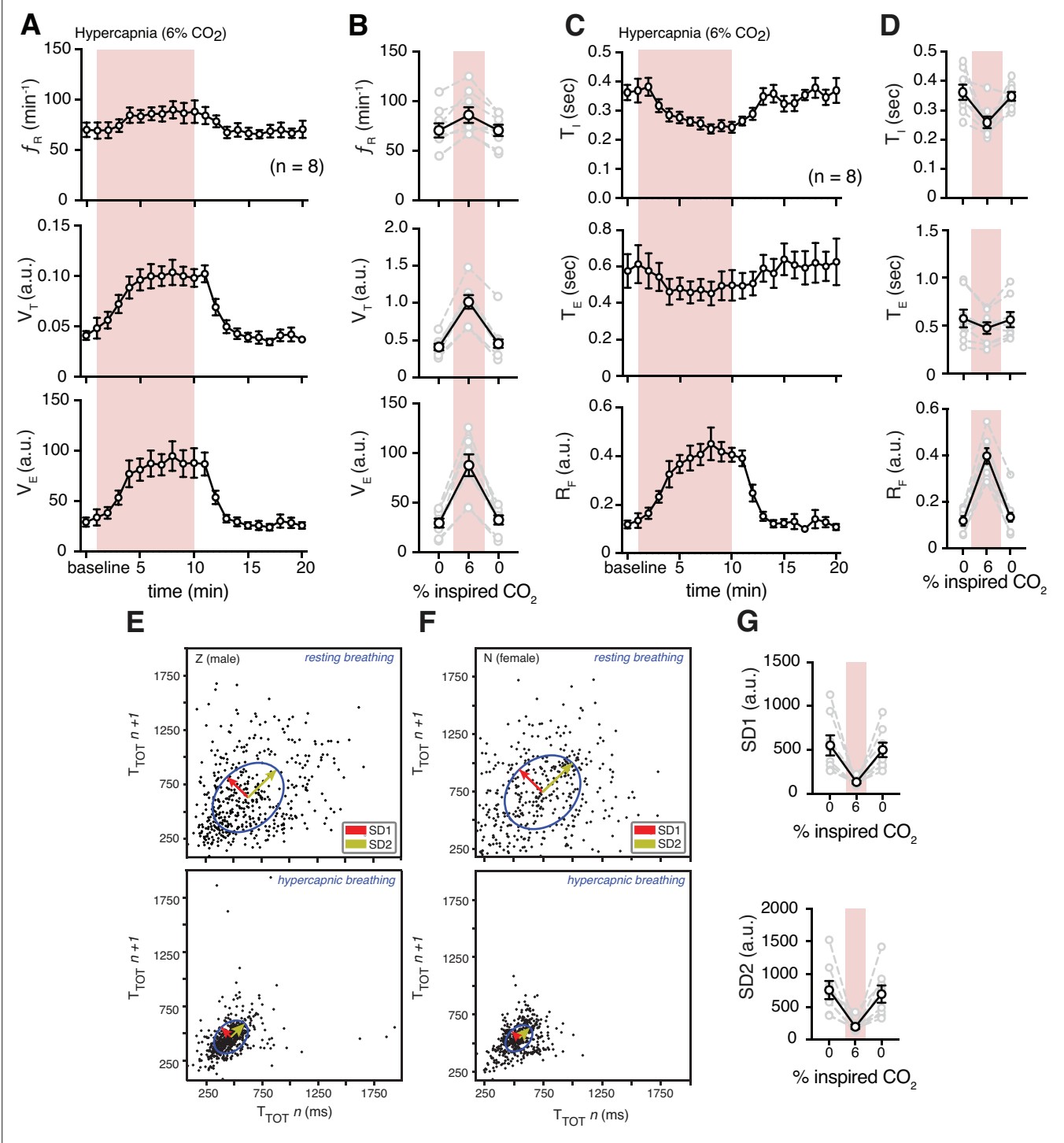

**Figure 3.** Hypercapnia challenge-induced changes in respiratory features. (**A**) Measurements of breathing rate ($f_R$), tidal volume ($V_T$), and minute ventilation ($V_E$) were averaged across 1 min epochs for assessment of local changes in each parameter. (**B**) Summaries of each feature at baseline, following 5 min exposure to hypercapnia, and in the 5 min immediately following the end of challenge. We observed increases in respiratory frequency ($p = 0.023$, Wilcoxon matched-pairs signed rank test), $V_T$ ($p = 0.008$, Wilcoxon matched-pairs signed rank test), and $V_E$ ($p = 0.008$, Wilcoxon matched-pairs signed rank test) during hypercapnia. (**C**) Measurements of inspiratory time ($T_I$), expiratory time ($T_E$), and respiratory drive ($R_D$) were averaged across 1 min epochs for assessment of local changes in each parameter. (**D**) Summaries of each feature at baseline (0% inspired $CO_2$), following 5 min exposure to 6% hypercapnia, and in the first 5 min following the end of hypercapnic challenge. During hypercapnia, we observed decreases in $T_I$ ($p = 0.008$, Wilcoxon matched-pairs signed rank test), $T_E$ ($p = 0.078$, Wilcoxon matched-pairs signed rank test), and increase in $R_D$ ($p = 0.008$, Wilcoxon matched-pairs signed rank test). Representative Poincaré plots of total cycle duration ($T_{TOT}$) for the nth cycle vs. $T_{TOT}$ for the nth+1 cycle during baseline (room air)

*Figure 3 continued on next page*

*Figure 3 continued*

and hypercapnic (6% $CO_2$) conditions in male (**E**) and female (**F**) marmosets. (**G**) Grouped data illustrating changes in SD1 and SD2 before, during, and after hypercapnia challenge. Respiratory rate variability decreased in both measures during hypercapnia compared to baseline (SD1: p = 0.008; SD2: p = 0.008; Wilcoxon matched-pairs signed rank test). In B, D, and G, data are shown as individual (gray lines) and mean values ± SEM (black line). a. u. – arbitrary unit.

The online version of this article includes the following source data and figure supplement(s) for figure 3:

**Source data 1.** Hypercapnia challenge source data.

**Figure supplement 1.** Hypercapnia challenge-induced changes in respiratory behavior by sex.

**Figure supplement 2.** Hypercapnia challenge-induced changes in respiratory features by sex.

**Figure supplement 3.** Changes in variability of respiration during hypercapnic challenge.

conditions did not elicit overall changes in $f_R$ (74 ± 5 vs. 82 ± 7 breaths $min^{-1}$ in baseline, p = 0.3, Wilcoxon matched-pairs signed rank test), but decreased $V_T$ (0.39 ± 0.08 vs. 0.54 ± 0.11 a.u. in baseline, p = 0.078, Wilcoxon matched-pairs signed rank test) and $V_E$ (29 ± 6 vs. 46 ± 12 a.u. in baseline, p = 0.043, paired *t* test) (*Figure 4*).

Then, we also calculated changes in $T_I$, $T_E$, and $R_F$ during hypoxic challenge with respect to baseline. Since $T_I$, $T_E$, and $R_F$ were not different in females and males during hypoxia (n = 4 per sex) (*Figure 4—figure supplement 2*), we combined their data. While hypoxia did not change $T_I$ (0.34 ± 0.02 vs. 0.30 ± 0.02 s in baseline, p = 0.46, Wilcoxon matched-pairs signed rank test) and $T_E$ (0.55 ± 0.1 vs. 0.50 ± 0.1 s in baseline, p = 0.4, Wilcoxon matched-pairs signed rank test), $R_F$ was decreased during hypoxia (12 ± 2 vs. 16 ± 3 a.u. in baseline, p = 0.008, Wilcoxon matched-pairs signed rank test) after 5 min of challenge (*Figure 4C and D*).

We also measured the effects of acute hypoxia on the regularity of breathing (*Figure 4E–G*). We combined the data from male and female marmosets as there were no sex differences measured for irregularity of breathing (*Figure 4—figure supplement 3*). We quantified the regularity of breathing by generating Poincaré plots and measuring SD1 and SD2. The baselines for SD1 and SD2 were similar during hypoxia (330 ± 39 vs. 374 ± 42 in baseline, p = 0.4, and 419 ± 52 vs. 488 ± 68 in baseline, p = 0.6, respectively; Wilcoxon matched-pairs signed rank test) (*Figure 4G*).

We then measured changes in respiratory features in the 5 min immediately following the hypoxic challenge (post-hypoxic challenge). Though we saw no changes in $f_R$ (73 ± 4 vs. 82 ± 7 breaths $min^{-1}$ in baseline, p = 0.11, Wilcoxon matched-pairs signed rank test) and $V_T$ (0.47 ± .10 vs. 0.54 ± 0.12 a.u. in baseline p = 0.15, Wilcoxon matched-pairs signed rank test), $V_E$ decreased (34 ± 7 vs. 46 ± 12 a.u. in baseline, p = 0.078, Wilcoxon matched-pairs signed rank test) relative to baseline (*Figure 4A and B*). We also calculated changes in $T_I$, $T_E$, and $R_F$, immediately following the hypoxia challenge. While we observed no change in $T_E$ (5.4 ± 0.5 vs. 5.0 ± 0.6 a.u. in baseline, p = 0.46), there was an increase in $T_I$ (3.4 ± 0.2 vs. 3.0 ± 0.2 a.u. in baseline, p = 0.055) and a decrease in $R_F$ (1.3 ± 0.2 vs. 1.8 ± 0.4 a.u. in baseline, p = 0.078) after hypoxia challenge (*Figure 4C and D*).

The constant $f_R$ and decrease in $R_F$ during hypoxia and post-hypoxic challenge suggests that the metabolic rate might decrease during acute hypoxic challenge. We then calculated the metabolic rate ($M_R$) in marmosets during hypoxic challenge. Our data suggest that $M_R$ had a profound decrease (~50%) during hypoxia when compared to the baseline (*Table 1*). Therefore, we calculated ventilatory efficiency as $V_E/M_R$ to understand the changes in ventilation in response to $CO_2$ production. Our analysis suggested that the ventilatory efficiency during hypoxic challenge was not different from the baseline (22 ± 6 vs. 19 ± 6 a.u. in baseline, p = 0.6, Wilcoxon matched-pairs signed rank test), however it was lower during post-hypoxic challenge (13 ± 3 vs. 22 ± 6 a.u. in hypoxia, p = 0.055, Wilcoxon matched-pairs signed rank test) (*Figure 4H*).

## Sigh frequency, sniffing, and apnea index in adult marmosets

Since incidences of sighs, apneas, and sniffing could contribute to the irregularity of respiration, we measured the frequencies of these essential features of breathing behavior. Sighs can be generated within the inspiratory rhythm-generating circuits of the preBötzinger complex (preBötC) (*Sheikhbahaei et al., 2018; Li et al., 2016; Lieske et al., 2000; Borrus et al., 2020; Toporikova et al., 2015; Vlemincx et al., 2013*), and may be modulated by excitatory signals from central chemocenters (*Sheikhbahaei et al., 2018; Sheikhbahaei et al., 2017; Souza et al., 2018; Souza et al., 2019;*

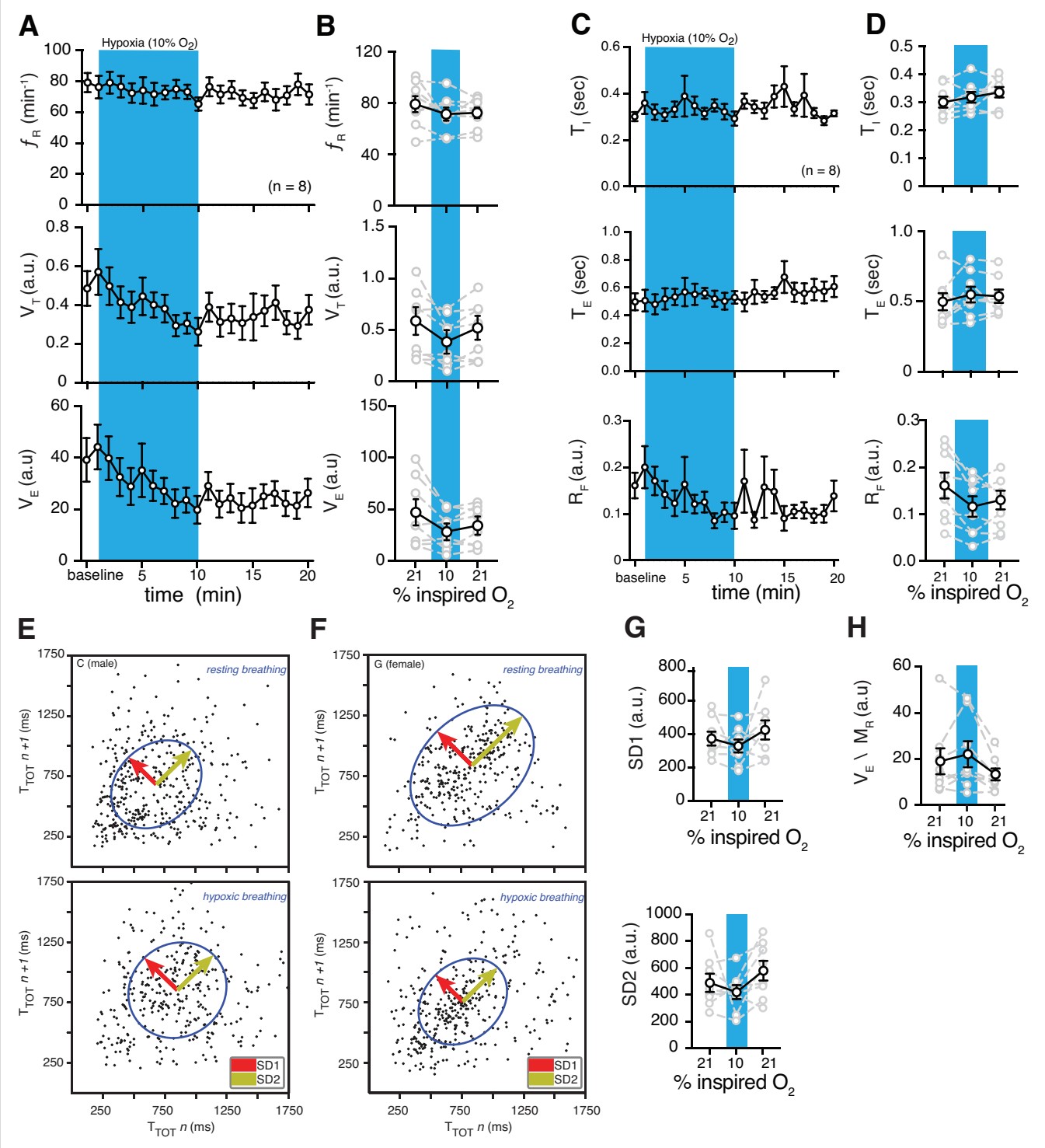

**Figure 4.** Hypoxic and post-hypoxic challenge-induced changes in respiratory features. (**A**) Measurements of breathing rate ($f_R$), tidal volume ($V_T$), and minute ventilation ($V_E$) were averaged across 1-min epochs for assessment of local changes in each parameter. (**B**) Summaries of each feature at baseline, following 5 min exposure to hypoxic (10% $O_2$) challenge, and in the 5 min immediately following the end of challenge. During hypoxia challenge, we saw no changes in respiratory frequency (p = 0.31, Wilcoxon matched-pairs signed rank test) and $V_E$ (p = 0.11, Wilcoxon matched-pairs signed rank test) compared to baseline. $V_T$ decreased during hypoxia challenge (p = 0.078, Wilcoxon matched-pairs signed rank test). Immediately following the challenge, we saw no changes in respiratory frequency (p = 0.11, Wilcoxon matched-pairs signed rank test) compared to baseline, and a post-challenge decrease in $V_T$ (p = 0.078, Wilcoxon matched-pairs signed rank test) and $V_E$ (p = 0.078, Wilcoxon matched-pairs signed rank test) compared to baseline. (**C**) Measurements of inspiratory time ($T_I$), expiratory time ($T_E$), and respiratory drive ($R_D$) were averaged across 1 min epochs for assessment of local

*Figure 4 continued on next page*

*Figure 4 continued*

changes in each parameter. (**D**) Summaries of each feature at baseline, following 5 min exposure to challenge until end of challenge, and in the 5 min immediately following the end of challenge. During hypoxic challenge, we saw no changes in respiratory $T_I$ (p = 0.5) or $T_E$ (p = 0.4), but a decrease in $R_D$ during (p = 0.008, Wilcoxon matched-pairs signed rank test) compared to baseline. We did observe post-hypoxic challenge increase in $T_I$ (p = 0.055) and $R_D$ (p = 0.023) and no change in $T_E$ (p = 0.46, Wilcoxon matched-pairs signed rank test). Representative Poincaré plots of total cycle duration ($T_{TOT}$) for the nth cycle vs. $T_{TOT}$ for the nth+1 cycle during baseline and hypoxic conditions (10% $O_2$) in male (**E**) and female (**F**) marmosets. (**G**) Summary data illustrating changes in SD1 and SD2 before, during, and after hypoxic challenge. Respiratory rate variability did not change for either measure during (SD1: p = 0.4; SD2: p = 0.6; Wilcoxon matched-pairs signed rank test) or after (SD1: p = 0.6; SD2: p = 0.3; Wilcoxon matched-pairs signed rank test) the hypoxic challenge compared to baseline. (**H**) Group data illustrating changes in ventilatory efficiency ($V_E/M_R$) before, during, and after hypoxic challenge. Ventilatory efficiency was not affected by acute hypoxia (p = 0.6, Wilcoxon matched-pairs signed rank test), however, it was lower during the post-hypoxic challenge (p = 0.055, Wilcoxon matched-pairs signed rank test). In B, D, G, and H, data are shown as individual (gray lines) and mean values ± SEM (black line). a. u. – arbitrary unit.

The online version of this article includes the following source data and figure supplement(s) for figure 4:

**Source data 1.** Hypoxia challenge source data.

**Figure supplement 1.** Hypoxic challenge-induced changes in respiratory behavior by sex.

**Figure supplement 2.** Hypoxic challenge-induced changes in respiratory features by sex.

**Figure supplement 3.** Changes in variability of respiration during hypoxic challenge.

*Li et al., 2016*). In female adult marmosets, sigh frequencies were not different when compared to those in male animals during the baseline in room air (11 ± 1 vs. 12 ± 2 hr$^{-1}$ in male) (*Figure 5—figure supplement 1*). In rodents, both hypoxic and hypercapnic challenges increased frequency of sighs (*Li et al., 2016*; *Sheikhbahaei et al., 2018*). Consistent with those results, hypoxia increased sigh events by 5.5 folds in marmosets (71 ± 10 vs. 11 ± 1 hr$^{-1}$ in room air, p = 0.008, Wilcoxon matched-pairs signed rank test). Similarly, hypercapnia also increased sigh frequency (68 ± 3 vs. 12 ± 1 hr$^{-1}$ in room air; p = 0.008, Wilcoxon matched-pairs signed rank test) (*Figure 5A*).

We also analyzed high-frequency breathing (sniffing) in marmosets. During the hypoxic challenge, the sniffing rate did not change with respect to baseline (74 ± 13 vs. 101 ± 38 hr$^{-1}$ in baseline, p = 0.84, Wilcoxon matched-pairs signed rank test) (*Figure 5B*). However, during hypercapnic challenge, rate of sniffing was less than that in room air (27 ± 16 vs. 100 ± 38 hr$^{-1}$ in baseline, p = 0.078, Wilcoxon matched-pairs signed rank test) (*Figure 5B*).

Spontaneous and post-sigh apneas have been reported in rodents, rabbits, humans, and other animals (*Yamauchi et al., 2008*; *Franco et al., 2003*; *van der Heijden and Zoghbi, 2018*; *Bongianni et al., 2010*; *Li et al., 2006*; *Ramirez et al., 2013*; *Sheikhbahaei et al., 2017*). We did not find differences in the apnea index between female and male marmosets (*Figure 5—figure supplement 1*). Apneas decreased during hypoxic challenge (37 ± 12 vs. 79 ± 20 in room air, p = 0.039, Wilcoxon matched-pairs signed rank test) (*Figure 5C*). During hypercapnic challenge, rate of spontaneous apneas also decreased drastically relative to that in room air (9 ± 6 vs. 129 ± 31 hr$^{-1}$ in room air, p = 0.008, Wilcoxon matched-pairs signed rank test) (*Figure 5C*).

## Spontaneous activity of adult marmosets

Lastly, to understand if hypoxia or hypercapnia have any effect on the animal's activity, we measured the movement of marmosets in the plethysmograph during both challenges. In measurements of large changes in position from one quadrant of the chamber to another, we saw no changes during hypoxia (4.1 ± 1.8 vs. 5.4 ± 1.6 quadrant changes per minute at baseline, n = 3, p = 0.99, Wilcoxon matched-pairs signed rank test) or hypercapnia (4.0 ± 0.5 vs. 4.7 ± 1.4 quadrant changes per minute at baseline, n = 3, p = 0.75, Wilcoxon matched-pairs signed rank test). Similarly, we observed no differences in the sum of frame-to-frame Euclidean distances in hypoxic challenge (124 ± 28 vs. 128 ± 27 pixels min$^{-1}$ at baseline, n = 3, p = 0.99, Wilcoxon matched-pairs signed rank test) or hypercapnia challenge (126 ± 20 vs. 120 ± 11 min$^{-1}$ at baseline, p = 0.99, Wilcoxon matched-pairs signed rank test) (*Figure 6*).

**Table 1.** Hypoxia decreased metabolic rate in common marmoset.

| | Pre-hypoxia | Hypoxia | Post-hypoxia |
|---|---|---|---|
| *Metabolic rate (%)* | 100 | 51 ± 4 | 98 ± 1 |

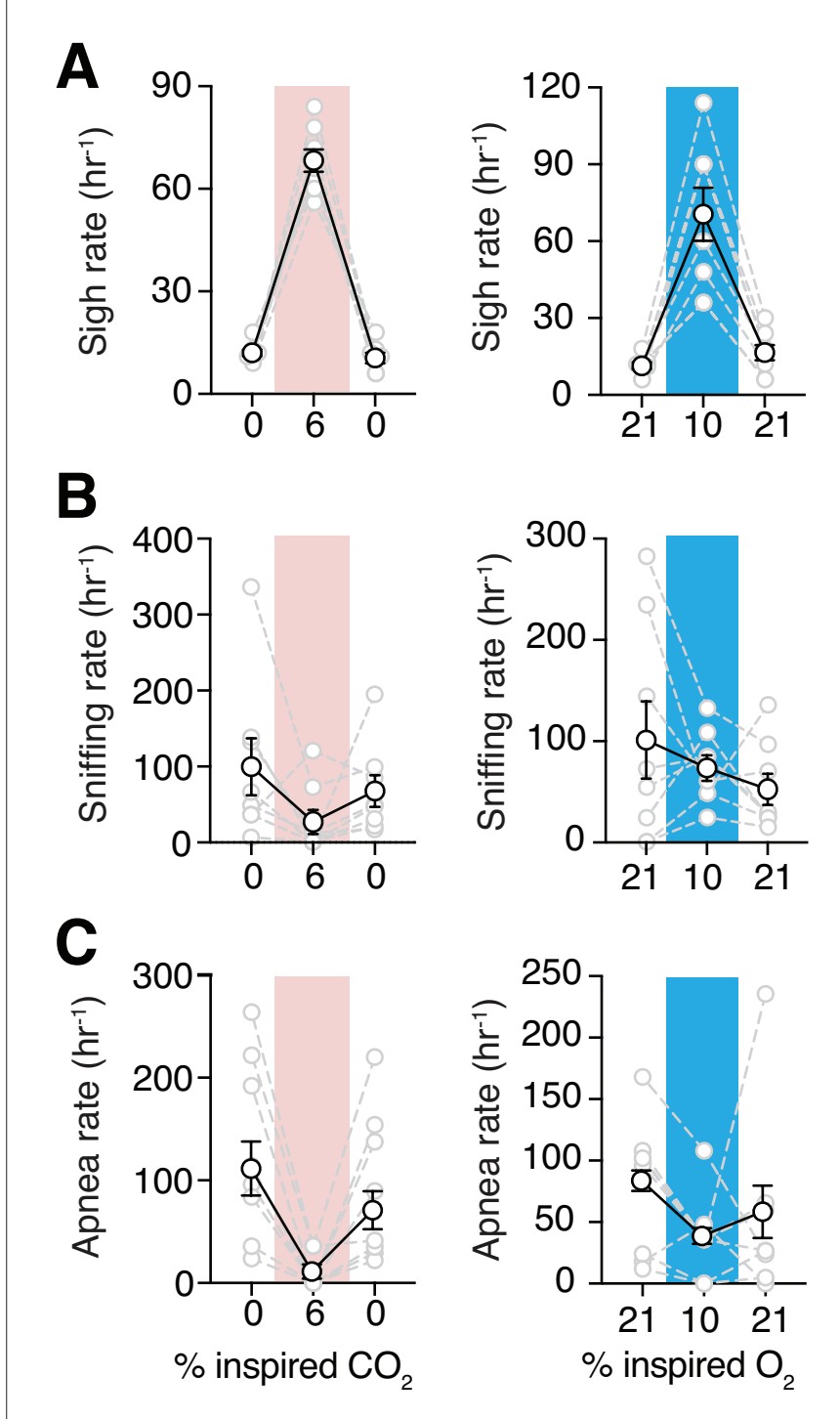

**Figure 5.** Sigh frequencies, sniffing rate, and apnea index during hypoxia and hypercapnia challenges.
(**A**) Summary data demonstrating increase in sigh frequency after 5 min of hypoxic (10% $O_2$, *left*) or hypercapnic
(6% $CO_2$, *right*) challenge (p = 0.008 and p = 0.008, respectively; Wilcoxon matched-pairs signed rank test).
(**B**) Summary data demonstrating no change in sniffing rate during (p = 0.74) and after (p = 0.74) hypoxia challenge
(*left*). Sniffing rate increased during and returned to baseline after hypercapnia challenge (p = 0.008 and p = 0.08
respectively; Wilcoxon matched-pairs signed rank test) (*right*). (**C**) Grouped data demonstrating a decrease in rate
of spontaneous apneas during hypoxia (p = 0.04, Wilcoxon matched-pairs signed rank test) (*left*) and hypercapnia
(p = 0.008, Wilcoxon matched-pairs signed rank test) (*right*). Data are shown as individual (gray lines) and mean
(black line) values ± SEM.

*Figure 5 continued on next page*

## Discussion

We used non-invasive, whole-body plethysmography to measure breathing behaviors (*Hamelmann et al., 1997*) in unrestrained, freely moving, awake marmosets, and rats. Plethysmography has a simple and robust design that has been used widely in humans (neonates [*Sivieri et al., 2017*] and adults [*Dubois et al., 1956*]), NHPs (such as macaques [*Besch et al., 1996*] and cynomolgus monkeys

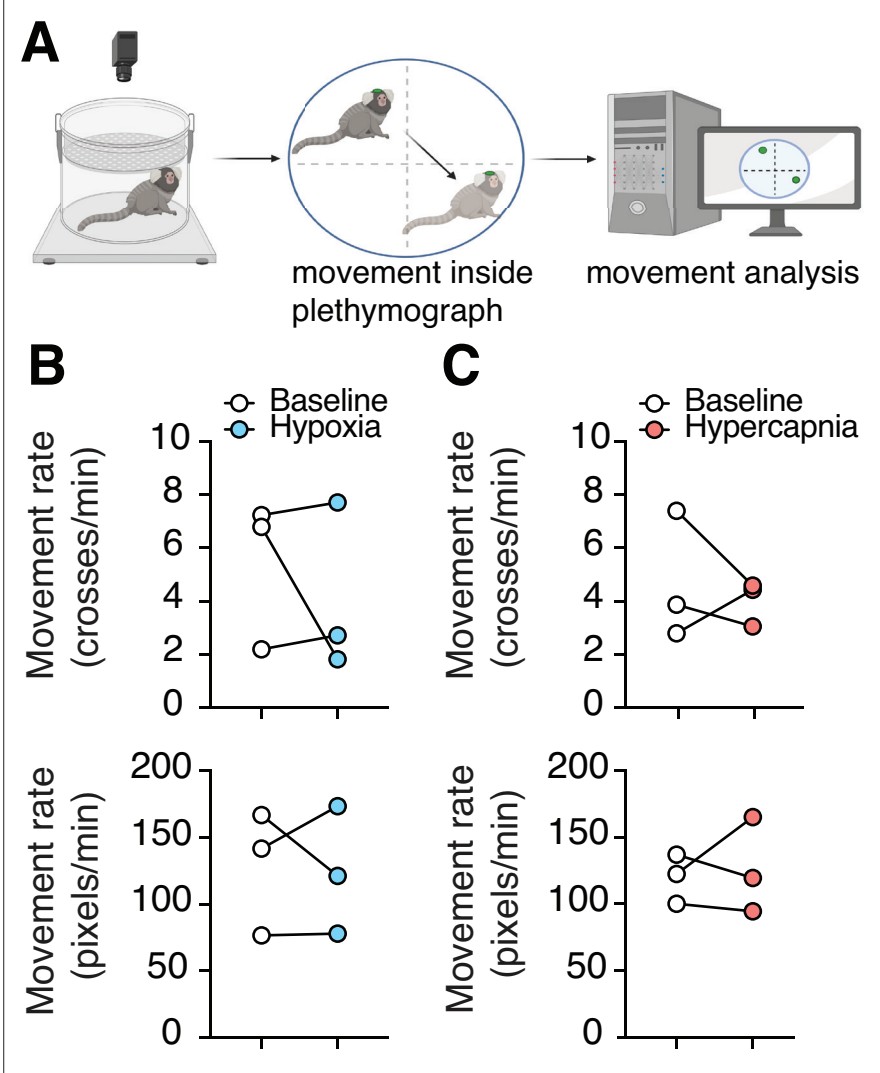

**Figure 6.** Changes in spontaneous activity during hypoxic and hypercapnic challenges. (**A**) Experimental design to analyze subject movement at baseline and during challenge. (**B**) Hypoxia did not induce any changes in animal movement rate as measured by quadrant changes in the chamber (top) (p = 0.99, n = 3; Wilcoxon matched-pairs signed rank test), or as measured by total change in animal position per second (bottom) (p = 0.99, n = 3; Wilcoxon matched-pairs signed rank test). (**C**) We detected no changes in animal's movement rate as measured by quadrant changes in the chamber (top) (p = 0.75, n = 3; Wilcoxon matched-pairs signed rank test), or by total change in position per second (bottom) (p = 0.99, n = 3; Wilcoxon matched-pairs signed rank test) during hypercapnia.

The online version of this article includes the following source data for figure 6:

**Source data 1.** Spontaneous activity source data.

[*Iizuka et al., 2010*]), rodents (*Sheikhbahaei et al., 2018*; *Hosford et al., 2020*), dogs (*Liu et al., 2016*), sheep (*Hutchison et al., 1983*), cats (*Hoffman et al., 1999*), turtles (*Valente et al., 2012*), and other animals.

However, analyzing whole-body respiratory data in conscious, awake animals requires complex algorithms to differentiate the respiratory signals. These respiratory data are commonly analyzed manually (or with proprietary software), therefore the analysis could be subjective, time-consuming, expensive, and/or not reproducible. To overcome this hurdle, we wrote a user-friendly, open-source Python script using Neurokit2, NumPy, and Pandas software packages (*McKinney, 2010*; *van der Walt et al., 2011*; *Makowski et al., 2020*) to analyze breathing behaviors from awake animal models. We then used our analysis tool to characterize breathing behaviors of awake common marmosets (*C. jacchus*) in their natural posture at rest, as well as during exposures to acute hypoxic and hypercapnic conditions.

The common marmoset is a small New World primate (*Okano et al., 2015*). Recently, marmosets have been proposed as a powerful animal model in neuroscience research (*Miller et al., 2016*; *Burkart and Finkenwirth, 2015*; *Leopold et al., 2017*; *Mitchell and Leopold, 2015*), especially to study vocal communication (*Eliades and Miller, 2017*). Compared to rodents, marmosets' central nervous system more closely resemble humans' in terms of physiological function and anatomy of the brain (*Bendor and Wang, 2005*). In addition, considering the similarity of the brain structure and circuit connectivity between primates, marmosets provide an attractive opportunity to study cortical (i.e., voluntary) control of motor activity (*Walker et al., 2017*). Furthermore, marmosets offer promise in understanding the coordination of breathing with complex behaviors, such as vocalization. However, the basic characteristics of breathing behaviors in the common marmoset had not been defined prior to this work.

## The ventilatory response to acute hypercapnia

Currently, the chemosensitivity mechanisms that adjust breathing with respect to the level of $PCO_2$/pH in the brain are centered around neurons and astrocytes in the retrotrapezoid nucleus (RTN) and medullary raphé (*Kumar et al., 2015*; *Teran et al., 2014*; *Guyenet et al., 2019*; *Gourine et al., 2010*). However, other data support a hypothesis that distributed chemosensitive regions in the medulla act as central respiratory chemosensors and are responsible for mounting of about 70% of the hypercapnic respiratory response (the mechanism that adjusts breathing in accordance with increase in $PCO_2$) (*Nattie, 1999*; *Nattie, 2000*; *Nattie, 2001*; *Spyer and Thomas, 2000*; *Nattie and Li, 2009*). Specialized peripheral chemoreceptors located in the carotid bodies (and aortic bodies in some species) are responsible for the remaining 30% of hypercapnia-induced augmentation of breathing. In awake, freely behaving marmosets, hypercapnia increased both breathing rate ($f_R$) and tidal volume ($V_T$) (*Figure 3*). However, the augmentation of ventilation ($V_E$) was mainly due to increase in $V_T$ (by ~160%) rather than $f_R$. These data are comparable to data obtained from rodents (*Bhandare et al., 2020*; *Sheikhbahaei et al., 2018*) and human (*Duffin et al., 2000*; *Ogoh et al., 2009*; *Serebrovskaya, 1992*; *Maxwell et al., 1986*). In this study we used hyperoxic hypercapnia. In humans and rodents, hyperoxia is proposed to suppress the activity of carotid bodies (*Chavez-Valdez et al., 2012*; *Gonzalez et al., 1994*; *Bates et al., 2014*). By extrapolation, since marmosets lack aortic bodies (*Clarke and de B Daly, 2002*), we assumed that hyperoxia attenuates marmoset's carotid body activity, and therefore, the hypercapnic ventilatory response presented here may represent the central $CO_2$ chemosensitive activity. Hypercapnia also increases frequency of sighs (i.e., augmented breath) in rodents (*Forsberg et al., 2016*; *Ramirez, 2014*). Consistent with these data, we also found that hypercapnia increased sigh frequency in marmosets (*Figure 5A*). Nevertheless, our data suggest that the common marmoset is a good animal model for studying respiratory responses to hypercapnia. However, more experiments are required to show that increases in $CO_2$ actually activate classical chemosensitive regions in marmosets.

## The ventilatory response to acute hypoxia

The hypoxic ventilatory response (HVR) in common marmosets was noteworthy, as there was little or no increase in $f_R$ during hypoxic exposure (*Figure 4A*). We believe the level of $O_2$ during hypoxia was sufficient to elicit HVR, as a similar level of $O_2$ (10% $O_2$) decreased the peripheral oxygen saturation ($SpO_2$) to 89% in humans after 180 s (*Gerlach et al., 2021*). In addition, increases in sigh rate and the

existence of post-hypoxic depression (see below) strongly suggest that the respiratory circuits were activated by the hypoxic challenge to prevent hypoxic ventilatory decline (HVD).

Although hypoxic conditions in marmosets' natural habitat (sea-level forests of the Amazon) are rare, hypoxia might occur as a result of disease or during sleep. Acute HVR is likely biphasic in mammals (*Easton et al., 1986*; *Rehan et al., 1996*; *Martin et al., 1998*; *Vizek et al., 1987*; *Waites et al., 1996*; *Gozal and Gaultier, 2001*; *Greer and Funk, 2013*). During acute hypoxia, ventilation shows an initial increase followed by a subsequent decline to a value at or above the baseline (i.e., HVD). This biphasic hypoxic response has been reported in humans, rats, and other mammals (*Eden and Hanson, 1987*; *Martin et al., 1990*; *Fung et al., 1996*; *Dahan et al., 1996*; *Vizek and Bonora, 1998*). However, earlier reports suggest that there is considerable interindividual variation in HVR in humans (*Hirshman et al., 1975*; *Weil and Zwillich, 1976*). Recent data in awake adult humans showed no increase of $f_R$ during acute hypoxia (*Gerlach et al., 2021*), suggesting that any changes in ventilation may be due to changes in $V_T$, not $f_R$ (*Tarbichi et al., 2003*). Our data support these reports, as we see variable responses to acute hypoxia in marmosets as well as a slight increase in $V_T$ and $V_E$ during the first minute of HVR followed by a decrease in $V_T$ and $V_E$ as hypoxia continues (*Figure 4*). However, ventilatory efficiency ($V_E /M_R$) was not affected by hypoxia (see below). It is possible that the large gas-exchange capacity of marmosets' lungs (due to the increased oxygen diffusion capacity) (*Barbier and Bachofen, 2000*) maintains the adequate blood oxygenation, and therefore, blunts the HVR during hypoxia.

Hypoxia increases sigh frequency in mammals, even in animals whose carotid bodies are non-functional (*Bartlett, 1971*; *Schwenke and Cragg, 2000*; *Cherniack et al., 1981*; *Sheikhbahaei et al., 2018*). Consistent with these data, hypoxia also increased sigh frequency in marmosets. In addition, the fact that sigh frequency, but not breathing frequency, increased during hypoxic challenge, supports the hypothesis that distinct cells may be responsible for the generation of rhythmic sighs and normal breathing (*Toporikova et al., 2015*; *Li et al., 2016*; *Sheikhbahaei et al., 2018*). Recent data from behaving rats suggest that purinergic signaling from astrocytes (numerous star-shaped glial cells) in the respiratory rhythm-generating circuits of the preBötC may play a significant role in regulation of sigh generation (*Sheikhbahaei et al., 2018*).

On the other hand, the mechanism of HVD is not fully understood. It is proposed that desensitization of peripheral chemoreceptors might play a role (*Bascom et al., 1990*), though significant evidence suggests that, at least in rodents, astrocytes in the preBötC are capable of acting as central respiratory oxygen chemosensors (*Sheikhbahaei et al., 2018*; *Angelova et al., 2015*; *Rajani et al., 2018*). Moreover, preBötC astrocytes might contribute to the HVD via release of adenosine triphosphate (ATP) (*Sheikhbahaei et al., 2018*). Existence of ATP receptors in the brainstem respiratory regions in marmosets (*Yao et al., 2000*) further strengthens this hypothesis in primates. In addition to the preBötC, RTN, rostral ventrolateral medulla, and the nucleus of the solitary tract in the brainstem are proposed to have oxygen sensing capabilities (*Accorsi-Mendonça et al., 2015*; *Mazza et al., 2000*; *Uchiyama et al., 2020*). However, more research is required to understand if this 'distributed central oxygen chemosensors' hypothesis (*SheikhBahaei, 2020*) can be generalized to primates.

Decrease of post-hypoxic ventilation in human is also reported (*Tarbichi et al., 2003*). This post-hypoxic depression is also illustrated in conscious (*Angelova et al., 2015*; *Sheikhbahaei et al., 2018*) and anesthetized (*Rajani et al., 2018*) rats. Similarly, we observe such a respiratory response in marmosets. However, we did not detect any sex differences during post-hypoxic recovery from hypoxia as reported in rat's in vitro models (*Garcia et al., 2013*). Our data are consistent with that reported in humans, namely that there are no differences in ventilation between sexes during post-hypoxic response (*Tarbichi et al., 2003*).

We also examined marmoset activity during hypoxic challenges. Our data, however, suggest that animal activity was not affected by hypoxia (or hypercapnia). This suggests that decreases in metabolic rate during hypoxic challenge are not due to decrease in spontaneous activity, but may be accounted for by changes in other processes with metabolic demand such as thermoregulation or cardiovascular activities.

Other than an increase in ventilatory response during hypoxia, mammals can reduce oxygen demand by optimizing and decreasing their rates of metabolism (*Dzal et al., 2015*). During acute hypoxia, adult marmosets decreased their metabolic rates ($M_R$) by ~50%, which is similar to data reported in other primates (pygmy marmosets [*Tattersall et al., 2002*] and humans [*Robinson and*

*Haymes, 1990*]), but two to three times more than the calculated rates from cats (*Gautier et al., 1989*) and rats (*Mortola et al., 1994*). This decrease in metabolism together with increase in sigh frequency might be sufficient for homeostatic control of blood oxygen during acute hypoxia in primates. We also analyzed ventilatory efficiency ($V_E/M_R$) to understand the changes in ventilation in response to $CO_2$ production. Although we saw a slight increase in ventilatory efficiency, acute hypoxia did not have a significant effect on $V_E/M_R$ (*Figure 4H*). We believe ventilatory efficiency gives a more comprehensive view on ventilation compared to just measuring $V_E$. It also suggests that acute hypoxia does not increase ventilation in the common marmoset.

We acknowledge that our characterization of breathing behaviors in the common marmoset is not complete and more experiments are needed to fully characterize hypoxic breathing behaviors in common marmosets. For instance, the respiratory response to hypoxia in rodents is non-linear, as a decrease in inspired $O_2$ to 15% elicits a minimal ventilatory response, but a decrease to 10% elicits a strongly robust one (*SheikhBahaei, 2017*; *Hosford et al., 2020*; *Sheikhbahaei et al., 2018*). In addition, a decrease in metabolic rate strongly suggests that the core body temperature is affected by hypoxia (*Morgan et al., 2014*). Since we did not measure core body temperature in the marmoset, we reported the tidal volume as arbitrary units. Future experiments (using telemetry probes or other devices) to measure core body temperature in marmosets are needed to accurately measure changes in tidal volume during stepwise changes of inspired $O_2$ or $CO_2$. However, our data suggest that the analysis toolbox presented in this study is a powerful means to analyze breathing data in awake animal models under different experimental $O_2$ and $CO_2$ conditions.

## Materials and methods
### Animals
We used 16 common marmosets (*C. jacchus*) (8 males, 8 females; 394 ± 5 g; 40 ± 1 months) and three male Sprague-Dawley rats (320 ± 11 g) for measuring and defining breathing behaviors. All experiments were performed in accordance with the National Institutes of Health Guide for the Care and Use of Laboratory Animals. The experiments on marmosets and rats were approved by the Animal Care and Use Committee (ACUC) of the Intramural Research Program (IRP) of the National Institute of Mental Health and ACUC of the IRP of National Institute of Neurological Disorders and Stroke, respectively. Animals were housed in temperature-controlled facilities on a normal light-dark cycle (12 hr:12 hr, lights on at 7:00 AM). They lived in paired or family-grouped housing and were given food and water ad libitum.

### Measurement of respiratory activity
Marmoset respiratory activity was measured using whole-body plethysmography in a room with ambient temperature of 27–28°C. Awake animals were placed in the Plexiglas chamber (~3 L) which was flushed with 21% $O_2$, 79% $N_2$, at a rate of 2.2 L min$^{-1}$ during measurements of baseline respiratory behavior (*Figure 1*). Concentrations of $O_2$ and $CO_2$ in the chamber were monitored using a fast-response $O_2/CO_2$ analyzer (ML206, AD Instruments). All experiments were performed at the same time of day (between 10:00 and 14:00 hr) to account for possible circadian changes in base level physiology (*Iizuka et al., 2010*). For measuring the respiratory behaviors during hypoxia, following a 40 min baseline period, the chamber was flushed with 10% $O_2$, 90% $N_2$, at a rate of 2.2 L min$^{-1}$. After 10 min of exposure to hypoxic conditions, the gas concentration in the chamber was changed to room air for another 10 min (*Figure 1—figure supplement 1*). Marmoset respiratory activity was also measured during exposure to hypercapnic conditions. Following a 40 min baseline period, the chamber was flushed with 6% $CO_2$, 60% $O_2$, 34% $N_2$, at a rate of 2.2 L min$^{-1}$. After 10 min of exposure to hypercapnic conditions, the chamber was then flushed with room air for another 10 min. Hyperoxic condition (60% $O_2$) was used to prevent any hypoxia associated with hypercapnia as used routinely in rodents (*Teppema et al., 1997*; *Sheikhbahaei et al., 2018*). Respiratory data were acquired with Power1401 (CED; RRID: SCR_017282) interface and transferred to Spike2 software (CED; RRID: SCR_000903). To prevent any acclimatization confound, each animal was placed only once in the plethysmography chamber and randomly assigned to either hypoxia or hypercapnia experiment.

Similarly, we used whole-body plethysmography to record respiratory activity in unrestrained conscious adult rats as described before (*Sheikhbahaei et al., 2017*). Briefly, adult rats were placed

in a Plexiglas recording chamber (~1 L) that was flushed continuously with room air (21% $O_2$, 79% $N_2$; temperature 22–24°C), at a rate of 1.2 L $min^{-1}$. The animals were allowed to acclimatize to the chamber environment for ~60 min. Resting breathing activity was then recorded for 10 min. Respiratory activity in all the animals was assessed at the same time of the day (between 10:00 AM and 2:00 PM) to take into the account circadian variations of the physiological parameters. Data were acquired using Power1401 interface and analyzed offline using either *Spike2* software (CED) or our in-house script presented in this paper.

## Calculation of metabolic rate

For measuring metabolic rate ($M_R$) in marmosets, we calculated $CO_2$ production using the following equation and expressed as percent: $M_R = \Delta CO_2 \times F_R/body\ mass$, where $\Delta CO_2$ is the peak changes in the $[CO_2]$ in the chamber as measured by the gas analyzer. $F_R$ is the flow rate through the plethysmography chamber (i.e., 2.2 L $min^{-1}$), and body mass is marmoset body mass (g).

## Automated quantification of marmoset activity

We tracked 10 points on the marmoset head and body (n = 3 animal per challenge) from an overhead view of the plethysmograph using WhiteMatter e3Vision cameras (e3Vision camera; e3Vision hub; White Matter LLC). We used DeepLabCut version 2.10.2 for pose estimation of these features (*Nath et al., 2019*; *Mathis et al., 2018*). We labeled 656 total frames from 16, 20–30 min videos recorded at 60 fps (95% was used for model training). We used ResNet-50-based neural network with default parameters for four iterations with five shuffles, and the test error was: 29.8 pixels, train: 2.4 pixels, with 0.6 p-cutoff, test error was: 14.0 pixels, train: 2.4 pixels (image size 600 by 800 pixels).

Below-threshold feature coordinates were then filled using methods from the B-SOiD Python toolkit (*Hsu and Yttri, 2021*). We used the average position of five points on the head for further analysis after qualitative assessment of consistent labeling accuracy. By dividing the labeled images in quadrants along the X- and Y-axes (X = 400 pixels, Y = 300 pixels), we counted the number of times large changes in position (i.e., movement) occurred. Quadrant positions were down-sampled to 2 s to avoid counting quadrant changes from when the animal paused near the dividing lines. Additionally, successive Euclidean distances were calculated for each point across each frame of the videos to produce total movement. Total linear distance was then divided by length of condition in minutes to obtain rate of activity in each condition.

## Respiratory data analysis

All animals in the study were included in the analysis. Plethysmography data were imported to Python using Neo Python package (*Van Rossum and Drake, 2011*; *Garcia et al., 2014*). We wrote a custom Python script using methods from Neurokit2, NumPy, and Pandas software packages (*McKinney, 2010*; *van der Walt et al., 2011*; *Makowski et al., 2020*). Areas of the signal with frequencies above 300 cycles per minute (~3.3 Hz) were excluded from analysis, as they were likely artifact resulting from movement inside the chamber. To ensure that we captured the full change in ventilation, we used steady-state responses to hypoxia and hypercapnia and analyzed the data 5 min after the start of each challenge. Neurokit2 methods were used for signal cleaning and extraction of instantaneous frequency, $T_{TOT}$ (total time of breath), $T_I$ (time of inspiration), $T_E$ (time of expiration), and amplitude (i.e., tidal volume [$V_T$]) from trough to peak of the signals (see *Figure 2*). The calculated $V_T$ was normalized to the body mass (g) of each animal. Mean inspiratory flow rate ($R_F$) was defined as the ratio of $V_T$ to $T_I$ ($V_T/T_I$). During hypoxia and hypercapnia challenges, the respiratory signals were analyzed in 1 min epochs to consider local changes in respiration parameters.

High-frequency breathing (i.e., sniffing) was defined as any breathing frequency between 250 cycles (2.5 Hz) and 300 cycles per minute. Apneas were defined by breathing cycles with $T_{TOT}$ greater than three times the average for each animal. Augmented breaths (i.e., sighs) were readily identifiable by using the criteria described in rats (*Sheikhbahaei et al., 2018*; *Sheikhbahaei et al., 2017*) and measured during the baseline and experimental conditions.

Two measures of rate variability were also calculated as described elsewhere (*Soni and Muniyandi, 2019*). SD1 is a measure of dispersion of $T_{TOT}$ perpendicular to the line of identity in the Poincaré plots, therefore demonstrating short-term variability. SD2 is a measure of dispersion of $T_{TOT}$ along the line of identity in the Poincaré plots, demonstrating long-term variability in respiratory rate.

SD1 and SD2 are calculated by:
$$SD_1{}^2 = \tfrac{1}{2} SDSD^2$$
$$SD_2{}^2 = 2SDT_{TOT}{}^2 - \tfrac{1}{2} SDT_{TOT}{}^2$$

where SD is the standard deviation of successive differences in $T_{TOT}$ and $SDT_{TOT}$ is the standard deviation in $T_{TOT}$.

All data were tested with Shapiro-Wilk test for normality and statistically compared by $t$ test, Wilcoxon matched-pairs signed rank test, or Mann–Whitney $U$ rank test as appropriate in Prism 9 (GraphPad, Inc; RRID: SCR_002798). Data are reported as mean ± SEM.

## Acknowledgements

We are grateful for invaluable mentorships, supports, and discussions with Drs David Leopold, Yogita Chudasama, and Jeffrey Smith. We thank Dr Gregory Funk for valuable consultations. We also thank the NIH Library Writing Center for manuscript editing assistance and Biorender (Biorender.com) for figure one. This work was supported by the Intramural Research Program (IRP) of the National Institutes of Health, NINDS, and NIMH.

## Additional information

### Funding

| Funder | Grant reference number | Author |
| --- | --- | --- |
| Intramural Research Program of the National Institutes of Health, NINDS and NIMH | ZIA NS009420 | Shahriar SheikhBahaei |

The funders had no role in study design, data collection and interpretation, or the decision to submit the work for publication.

### Author contributions

Mitchell Bishop, Formal analysis, Methodology, Software, Visualization, Writing – original draft; Maximilian Weinhold, Formal analysis, Software, Validation, Visualization, Writing – review and editing; Ariana Z Turk, Afuh Adeck, Data curation, Writing – review and editing; Shahriar SheikhBahaei, Conceptualization, Data curation, Formal analysis, Funding acquisition, Investigation, Methodology, Project administration, Resources, Supervision, Validation, Visualization, Writing – original draft, Writing – review and editing

### Author ORCIDs

Mitchell Bishop http://orcid.org/0000-0003-4107-7015
Shahriar SheikhBahaei http://orcid.org/0000-0003-4119-9979

### Ethics

All experiments were performed in accordance with the National Institutes of Health Guide for the Care and Use of Laboratory Animals and were approved by the Animal Care and Use Committee of the Intramural Research Program of the National Institute of Mental Health and the Intramural Research Program of the National Institute of Neurological Disorders and Stroke.

### Decision letter and Author response

Decision letter https://doi.org/10.7554/eLife.71647.sa1
Author response https://doi.org/10.7554/eLife.71647.sa2

## Additional files

### Supplementary files
• Transparent reporting form

## Data availability

All the code is available on the NGSC GitHub (https://github.com/NGSC-NINDS/Marm_Breathing_Bishop_et_al_2021; copy archived at swh:1:rev:a1b78d7283653adc62a5ede1ea0b913ab5d1dd8a). The data generated in Figures 3–6 are provided in the source files.

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
