## [Editor Report]

The authors have thoughtfully revised their manuscript, with an increased focus on their breath analysis toolkit. We think this will be a tremendous resource for the respiratory community and we are hopeful it will decrease the barriers for others to conduct similar investigations.

---

## [Decision Letter]

**Decision letter after peer review:**

Thank you for submitting your article "Breathing Behaviors in Common Marmoset (*Callithrix jacchus*)" for consideration by *eLife*. Your article has been reviewed by 2 peer reviewers, and the evaluation has been overseen by a Reviewing Editor, Melissa Bates, and Michael Taffe as the Senior Editor. The reviewers have opted to remain anonymous.

Essential revisions:

1) Both reviewers point out that the major contributions of this study, particularly for people that are not experts in respiration, are not immediately clear. While Reviewer #1 saw the value in an automated system to quantify breathing, it is evident from Reviewer #2's comments that the value of this compared to current manual methods of quantification is not immediately apparent. It will likely also not be apparent to readers who have not toiled through manual breath analysis. As Reviewer #1 notes, these analysis tools are not widely available, so it is worth expanding on how analysis is typically performed versus how this new tool may enhance research in this field.

2) Reviewers #1 and 2 note questions about the marmoset model and both remark that it is unclear whether this is a paper that investigates similarities between non-human primates and humans, or a presentation of a new methodology. Reviewer #2 notes that the rationale for measuring ventilatory drive is not clear, nor whether the marmoset is a superior model to the rodent. Reviewer #1 notes serious concerns about some of the limitations to the hypoxic ventilatory drive measurements. It is possible that there is too much being done in this single paper and that this has tempered the reviewers' enthusiasm. Some of the limitations to the method of measuring the ventilatory response limit its ability to be compared to humans, as the marmoset changes its metabolic rate more like a rodent. In fact, this is a novel finding in a primate model but the paper may not benefit from having the characterization of the ventilatory drive as its central focus.

*Reviewer #1 (Recommendations for the authors):*

The absolute strongest contribution that this paper makes is the NeuroKil2 tool and the demonstration of its use in a novel research model. I think it would make the paper stronger to refocus the section on the hypoxic and hypercapnia responses as tools to provoke changes in ventilation without claiming to have completely characterized the response, given the limitations addressed in the public review.

There are also some generalizations that are made that are not necessarily true. For example, the authors claim in line 74 that rodents have not necessarily informed human disease, which is not true (see https://www.nejm.org/doi/full/10.1056/nejmc1311092 and https://journals.physiology.org/doi/full/10.1152/japplphysiol.00985.2002). The authors also discuss variability in the human ventilatory, response, and allude to the biphasic response. I did not understand the sentence in line 428 "In awake adult human, even one-hour hypoxic condition did not affect mean pulmonary minute ventilation (i.e., VE) (Robinson and Haymes 1990; Seo et al., 2017)." This implies that humans lack a hypoxic ventilatory response. Is this the authors' intent?

*Reviewer #2 (Recommendations for the authors):*

Below are specific comments to the individual parts of the text.

There are quite a few places where language needs corrections.

“In this study, we described breathing behaviors of common marmosets in a variety of O2 and CO2 conditions.”

As conclusion one wants to see a statement of novel findings and how they relate to the previous state of knowledge. What is the key finding of the paper?

“We also modified and optimized an analysis toolkit for automated characterization of respiratory indices in laboratory animals.”

This makes the scope of the paper unclear. Is it about some unexpected/interesting findings in NHP or is it a methodological paper to describe an evaluation protocol? If so, how do we know that it is any better than what was used before?

“All experiments were performed at the same time of day (between 10:00 and 14:00 hours)”

While the logic is clear it actually would have make the study much more exciting to see day/night comparisons

“To prevent any acclimatization confound, each animal was only placed once in the plethysmography chamber and randomly assigned to either hypoxia or hypercapnia experiment.”

But this raises a concern that some of the noted effects were induced by stress.

“We tracked 10 points on the marmoset head and body (n = 3 animal per challenge) from an overhead view of the plethysmograph using WhiteMatter e3Vision cameras (e3Vision camera; e3Vision hub; White Matter LLC). We used DeepLabCut version 2.10.2 for pose estimation of these features (Nath et al., 2019; Mathis et al., 2018). We abelled 656 total frames from 16, 20-30-minute videos recorded at 60 fps (95% was used for model training).”

The introduction does not prepare the reader to understand why you needed that.

“Total movement was then divided by length of condition in minutes to obtain rate of activity in each condition.”

Not clear what is means. Linear distance, vertical distance or combination?

“High frequency breathing (i.e., sniffing) was defined as any breathing frequencies between 250 cycles (2.5 Hz) and 300 cycles per minute.”

Given that the animals were on video would it not be more convincing to actually visually define sniffing and specifically analyse it?

“We did not identify any differences in values of ƒ216 R, VT, and VE when using Neurokit2 or conventional methods (n = 3) (Figure 2 —figure supplement 1).”

This comes as a disappointment because from here, one may ask, why did the authors spent time developing a different method if it gives the same result, what is the value of this undertaking?

“Then, we also calculate changes in time of inspiration (TI), time of expiration (TE), and respiratory drive (RD) during hypoxic challenge”

What is the meaning of this parameter, it is not even discussed or interpreted in any way.

“In measurements of large changes in position from one quadrant of the chamber to another, we saw no changes in hypoxia challenge (4.1 {plus minus} 1.8 vs 5.4 {plus minus} 1.6 quadrant changes per minute at baseline, n = 3, p = 0.99, Wilcoxon matched pairs signed rank test) or hypercapnia challenge”

One may wonder whether this way to detection is informative enough. There seems to be a general belief that in humans high CO2 is perceived as an alarm signal. Was there really no change in behaviour?

There are some questions here.

The authors write “In our experiments, to prevent hypoxia, we applied hyperoxic hypercapnia (60% O2/6% CO2 balanced with N2) as it is routinely used in rodent experiments”. Possibly I do not understand something here but why an increased CO2 which makes animals breathe harder should lead to hypoxia if the ambient level of O2 is maintained? Would it not rather lead to hyperoxia?

One would perhaps want to compensate by reducing N2 and do something like 6% CO2, 21% O2, balanced by N2. I wonder if the hyperoxia is without effect of its own. Then what are we looking at?

Line 395

“It has been shown that hyperoxia minimizes the input from peripheral chemosensors of carotid bodies (Chavez-Valdez et al., 2012; Gonzalez 396 et al., 1994). Since marmosets lack aortic bodies (Clarke and de B Daly 2002), the hypercapnic ventilatory response reported here is driven by the central CO2 respiratory chemocenters."

I may be wrong but don't think that the effect on hyperoxia on carotid bodies has ever been tested in primates, at least Chavez paper is in rats. I think there is a leap of faith here.

In the discussion in line 410 the authors say that there did not see increases in Vt but in the next paragraph (line 434) they say that they saw a slight increase in Vt. Which one is correct?

In relation to the mechanism behind reduction of the metabolic rate, the authors write

Line 481 "This suggests that decreases in metabolic rate during hypoxic challenge are not due to decrease in spontaneous activity, but may be accounted for by changes in other processes with metabolic demand such as thermoregulation"

I find it hard to believe that metabolic processes could adapt to lower oxygen so instantly and even less to, animals temperature. To cool down even a degree so quickly they would need to stop producing heat completely. I wonder whether the method used to assess metabolic rate is sufficiently fast/reliable

---

## [Author Response]

Essential revisions:1) Both reviewers point out that the major contributions of this study, particularly for people that are not experts in respiration, are not immediately clear. While Reviewer #1 saw the value in an automated system to quantify breathing, it is evident from Reviewer #2's comments that the value of this compared to current manual methods of quantification is not immediately apparent. It will likely also not be apparent to readers who have not toiled through manual breath analysis. As Reviewer #1 notes, these analysis tools are not widely available, so it is worth expanding on how analysis is typically performed versus how this new tool may enhance research in this field.

We acknowledge that the initial submission was not as clear as we had hoped. We have revised the manuscript and added more details about our new analysis script. We further demonstrated the applicability of the model by adding new analysis in rodents. We believe the major contribution of this manuscript to the field is providing an analysis tool to study complex breathing behavior in conscious, awake, and active laboratory animals. Thus, in this manuscript, we have applied and optimized our script on analysis breathing behaviors in common marmosets and rats.

2) Reviewers #1 and 2 note questions about the marmoset model and both remark that it is unclear whether this is a paper that investigates similarities between non-human primates and humans, or a presentation of a new methodology. Reviewer #2 notes that the rationale for measuring ventilatory drive is not clear, nor whether the marmoset is a superior model to the rodent. Reviewer #1 notes serious concerns about some of the limitations to the hypoxic ventilatory drive measurements. It is possible that there is too much being done in this single paper and that this has tempered the reviewers' enthusiasm. Some of the limitations to the method of measuring the ventilatory response limit its ability to be compared to humans, as the marmoset changes its metabolic rate more like a rodent. In fact, this is a novel finding in a primate model but the paper may not benefit from having the characterization of the ventilatory drive as its central focus.

In the revised manuscript, we put emphasis on the new methodology for analyzing breathing behaviors in awake laboratory animals. We believe it is difficult to assume one animal model to be *the best model for all research questions*! Although utilizing rodent models have some advantages, including availability of genetic tools, certainly having a primate model to study human physiological studies is more advantageous.

Because marmosets are primates, their brain circuits more closely resemble humans’, especially when compared to rodents. For instance, it is important to study cortical control of breathing, which is necessary for many voluntary activities, including speech/vocal production. This volitional control of breathing has not been shown in rodent models, and so marmosets, whose vocal production mirrors that of humans in numerous domains, would be the best fit for these types of studies. Therefore, it is advantageous to study the brain circuits involved in controlling respiratory behaviors in marmosets. In our opinion, designing necessary tools and characterizing basic marmoset breathing is the first and most important step in studying cortical control of breathing.

In terms of mirroring study methodology in humans and non-human primates, we believe that it is not a simple task. The experimental set ups are different when human subjects are being studied (i.e., using masks, whole-body plethysmography, subject laying down or sitting, etc.). These different methods certainly will affect the data. In addition, existing data on breathing behaviors between human and rodent models are not consistent (for example see PMID: 31445081).

Reviewer #1 (Recommendations for the authors):The absolute strongest contribution that this paper makes is the NeuroKil2 tool and the demonstration of its use in a novel research model. I think it would make the paper stronger to refocus the section on the hypoxic and hypercapnia responses as tools to provoke changes in ventilation without claiming to have completely characterized the response, given the limitations addressed in the public review.

We thank the reviewer for this comment. In the revised manuscript, we have shifted the focus to be on our analysis tool. We have also mentioned that we used ‘acute hypoxia/hypercapnia’ to perturb the ventilatory response, thus clarifying the findings and limitations of this study. In addition, we have removed any indication/claims that the breathing behaviors are *completely* characterized in marmosets, viewing this paper as a necessary but not sufficient contribution to the complete characterization of breathing behavior.

There are also some generalizations that are made that are not necessarily true. For example, the authors claim in line 74 that rodents have not necessarily informed human disease, which is not true (see https://www.nejm.org/doi/full/10.1056/nejmc1311092 and https://journals.physiology.org/doi/full/10.1152/japplphysiol.00985.2002).

We thank the reviewer for this comment. However, we believe there are other possible explanations for the studies provided by the reviewer. For example, many lines of evidence suggest that astrocytes in the ventrolateral medulla act as oxygen sensors and can modulate the activities of respiratory circuits (for instance see PMCID: PMC5785528; PMCID: PMC4510287; PMCID: PMC6068109). Moreover, matching rodent age to human age is questionable as rodents mature rapidly compared to humans. Therefore, we believe comparing animal studies to human studies is not as straightforward as we would hope. However, to respond to the comment of the reviewer, we have removed the sentence in line 74.

The authors also discuss variability in the human ventilatory, response, and allude to the biphasic response. I did not understand the sentence in line 428 "In awake adult human, even one-hour hypoxic condition did not affect mean pulmonary minute ventilation (i.e., VE) (Robinson and Haymes 1990; Seo et al., 2017)." This implies that humans lack a hypoxic ventilatory response. Is this the authors' intent?

Several studies showed variability of hypoxic ventilatory responses in humans (for instance see PMID: 1141125). Even in the paper provided by the reviewer (PMCID: PMC4769592, previous comment), there are considerable variations in hypoxic responses in both control and experimental subjects.

In line 428, Robinson and Haymes, applied hypoxia for 1.5 hrs and in the first 1 hour, there was no increase in V_E_. We do not believe that these data suggest that humans lack hypoxic ventilatory response, rather further points toward variability of responses (possibly due to different experimental methods). However, to prevent any confusion, we have removed the sentence from our revised manuscript.

Reviewer #2 (Recommendations for the authors):Below are specific comments to the individual parts of the text.52 In this study, we described breathing behaviors of common marmosets in a variety of O2 and CO2 conditions.As conclusion one wants to see a statement of novel findings and how they relate to the previous state of knowledge. What is the key finding of the paper?

Thank you for the comment. We have revised the abstract to capture the message of the paper in the revised manuscript.

“ We also modified and optimized an analysis toolkit for automated characterization of respiratory indices in laboratory animals.”This makes the scope of the paper unclear. Is it about some unexpected/interesting findings in NHP or is it a methodological paper to describe an evaluation protocol? If so, how do we know that it is any better than what was used before?

In the revised text of the manuscript, we have changed the focus to be on the development of the analysis toolkit. We have also added more information about this open-source tool to analyze breathing behaviors in animal models.

“All experiments were performed at the same time of day (between 10:00 and 14:00 hours)”While the logic is clear it actually would have make the study much more exciting to see day/night comparisons

Like humans, marmosets sleep during the night so we are not sure if measuring breathing behaviors during the night would give us more information. The advantage of our new tool in analysis of breathing behaviors is that it can analyze the breathing during the time that the animal is awake and active.

“To prevent any acclimatization confound, each animal was only placed once in the plethysmography chamber and randomly assigned to either hypoxia or hypercapnia experiment.”But this raises a concern that some of the noted effects were induced by stress.

We have waited 30-40 minutes for the breathing behaviors to reach a plateau. Most marmosets calm down in 20-30 minutes. Although we cannot rule out any stress involvement in our recordings (since we did not sample blood from animal), it is highly unlikely that the animals are stressed as they do not show any stress behavior/vocalization. It has to be noted that this acclimatization time is also sufficient for rodents to get used to the plethysmography chamber.

“ We tracked 10 points on the marmoset head and body (n = 3 animal per challenge) from an overhead view of the plethysmograph using WhiteMatter e3Vision cameras (e3Vision camera; e3Vision hub; White Matter LLC). We used DeepLabCut version 2.10.2 for pose estimation of these features (Nath et al., 2019; Mathis et al., 2018). We labeled 656 total frames from 16, 20-30-minute videos recorded at 60 fps (95% was used for model training).”The introduction does not prepare the reader to understand why you needed that.

Thank you for the comment. We have added more information in the result section about this analysis. Since analysis of animal movement is not central to the paper, we decided to not describe it in the introduction.

“Total movement was then divided by length of condition in minutes to obtain rate of activity in each condition.”Not clear what is means. Linear distance, vertical distance or combination?

Thank you for this comment. We have addressed the ambiguity in our explanation, as the successive differences mentioned in the previous line are calculated as linear distances.

“High frequency breathing (i.e., sniffing) was defined as any breathing frequencies between 250 cycles (2.5 Hz) and 300 cycles per minute.”Given that the animals were on video would it not be more convincing to actually visually define sniffing and specifically analyse it?

As opposed to rodents, sniffing is difficult to ‘see’ in marmoset. The power of our new analysis tool is in its ability to quantify all the respiratory behaviors in one package.

“We did not identify any differences in values of ƒ216 R, VT, and VE when using Neurokit2 or conventional methods (n = 3) (Figure 2 —figure supplement 1).”This comes as a disappointment because from here, one may ask, why did the authors spent time developing a different method if it gives the same result, what is the value of this undertaking?

We respectfully disagree. We bench marked our new analysis tool to the conventional method to show that it can analyze *the basic* characteristics of breathing (i.e., of ƒ_R_, V_T_, and V_E_) as good as the manual/conventional methods. We believe this validation step is necessary for every new analysis toolbox. In addition, our automated, open-source script can rapidly perform analysis, much faster than the traditional method of analyzing breathing behavior and analyze other breathing behaviors.

“Then, we also calculate changes in time of inspiration (TI), time of expiration (TE), and respiratory drive (RD) during hypoxic challenge”What is the meaning of this parameter, it is not even discussed or interpreted in any way/

In the revised manuscript, we have changed this variable to Respiratory Flow (R_F_) and described it in the method section. We have also discussed the R_F_ data in the main text.

“In measurements of large changes in position from one quadrant of the chamber to another, we saw no changes in hypoxia challenge (4.1 {plus minus} 1.8 vs 5.4 {plus minus} 1.6 quadrant changes per minute at baseline, n = 3, p = 0.99, Wilcoxon matched pairs signed rank test) or hypercapnia challenge”One may wonder whether this way to detection is informative enough. There seems to be a general belief that in humans high CO2 is perceived as an alarm signal. Was there really no change in behaviour?

There was no change in spontaneous movements of the animal during both hypoxia and hypercapnia. We also did not see any behaviors indicating stress of the marmosets during the hypercapnic challenge. However, it is possible that higher CO_2_ concentrations may elicit changes in marmoset’s behavior, but 6% was only affected the breathing behaviors.

There are some questions here.The authors write "In our experiments, to prevent hypoxia, we applied hyperoxic hypercapnia (60% O2/6% CO2 balanced with N2) as it is routinely used in rodent experiments". Possibly I do not understand something here but why an increased CO2 which makes animals breathe harder should lead to hypoxia if the ambient level of O2 is maintained? Would it not rather lead to hyperoxia?One would perhaps want to compensate by reducing N2 and do something like 6% CO2, 21% O2, balanced by N2. I wonder if the hyperoxia is without effect of its own. Then what are we looking at?

Thank you for the comment. In the revised text, we have fixed this error.

Line 395"It has been shown that hyperoxia minimizes the input from peripheral chemosensors of carotid bodies (Chavez-Valdez et al., 2012; Gonzalez 396 et al., 1994). Since marmosets lack aortic bodies (Clarke and de B Daly 2002), the hypercapnic ventilatory response reported here is driven by the central CO2 respiratory chemocenters."I may be wrong but don't think that the effect on hyperoxia on carotid bodies has ever been tested in primates, at least Chavez paper is in rats. I think there is a leap of faith here.

The effect of hyperoxia (60%) in baseline breathing is tested in humans (PMCID: PMC4769592) and it has been shown that hyperoxia depresses baseline *V*_E_ and the effect is suggested due to inhibition of carotid bodies. However, to respond to the comment of the reviewer, we have discussed this issue in the manuscript.

In the discussion in line 410 the authors say that there did not see increases in Vt but in the next paragraph (line 434) they say that they saw a slight increase in Vt. Which one is correct?

Thank you for the comment. These two V_T_s are not the same. In line 471, the text points toward the V_T_ in the first minute of hypoxic challenge, however, in line 472, the discussed V_T_ is the average tidal volume after 5 minutes of hypoxia. We now clearly stating this in the revised text.

In relation to the mechanism behind reduction of the metabolic rate, the authors writeLine 481 "This suggests that decreases in metabolic rate during hypoxic challenge are not due to decrease in spontaneous activity, but may be accounted for by changes in other processes with metabolic demand such as thermoregulation"I find it hard to believe that metabolic processes could adapt to lower oxygen so instantly and even less to, animals temperature. To cool down even a degree so quickly they would need to stop producing heat completely. I wonder whether the method used to assess metabolic rate is sufficiently fast/reliable

The calculated metabolic rate is not instantaneous and is done after five minutes of hypoxic challenge. In the revised text, we have clarified this issue.